



# The Detailed Emissions Scaling, Isolation, and Diagnostic (DESID) module in the Community Multiscale Air Quality (CMAQ) Modeling System version 5.3

Benjamin N. Murphy[1], Christopher G. Nolte[1], Fahim Sidi[1], Jesse O. Bash[1], K. Wyat Appel[1], Carey Jang[2],
Daiwen Kang[1], James Kelly[2], Rohit Mathur[1], Sergey Napelenok[1], George Pouliot[1] and Havala O. T. Pye[1]

[1]Center for Environmental Measurement and Modeling, U.S. Environmental Protection Agency, Research Triangle Park,
North Carolina, 27711, U.S.A.
[2]Office of Air Quality Planning and Standards, U.S. Environmental Protection Agency, Research Triangle Park, North
Carolina, 27711, U.S.A.

*Correspondence to*: Benjamin N. Murphy (murphy.ben@epa.gov)

**Abstract.** Air quality modeling for research and regulatory applications often involves executing many emissions sensitivity cases to quantify impacts of hypothetical scenarios, estimate source contributions or quantify uncertainties. Despite the prevalence of this task, conventional approaches for perturbing emissions in chemical transport models like the Community Multiscale Air Quality (CMAQ) model require extensive offline creation and finalization of alternative emissions input files. This workflow tends to be time-consuming, error-prone, inconsistent among model users and difficult to document while consuming increased computer storage space. The Detailed Emissions Scaling, Isolation, and Diagnostic (DESID) module, a component of CMAQv5.3 and beyond, addresses these limitations by performing these modifications online during the air quality simulation. Further, the model contains an Emission Control Interface which allows users to prescribe both simple and highly complex emissions scaling operations with control over individual or multiple chemical species, emissions sources, and spatial areas of interest. DESID further enhances the transparency of its operations with extensive error-checking and optional gridded output of processed emission fields. These new features are of high value to many air quality applications including routine perturbation studies, atmospheric chemistry research, and coupling with external models (e.g. energy system models, reduced-form models).

## 1 Introduction

Air pollution causes significant adverse health effects, including premature mortality, with more than 4 million deaths attributed to $PM_{2.5}$ (particulate matter with diameter less than 2.5 micrometers) and ozone exposure globally in 2015 (USEPA, 2019b; Cohen et al., 2017; Burnett et al., 2018). Governments around the world have made significant efforts to improve air quality to alleviate the harms caused by air pollution at multiple scales from near-source emissions (e.g. indoor heating and cooking, roadway, uncontrolled burning, industry and energy generation) to regional transport and production (secondary



ozone, secondary particulate matter). Chemical transport models (CTMs) provide the latest scientific representations of the key processes (emission, transport, reaction, and deposition) that govern pollutant concentrations and are used extensively by air quality managers in improving urban- to regional-scale air quality.

For air pollution management applications, these models are typically used to simulate recent periods of elevated pollutant
concentrations in a study region, using the best-available representation of the pollutant emissions, pollutant physicochemical properties and coincident meteorology that occurred. Model skill is quantified by evaluating predictions against observations using statistical metrics and generally accepted performance criteria (e.g., USEPA, 2018; Kelly et al., 2019; Emery et al., 2017; Simon et al., 2012). Once acceptable model performance is demonstrated, air quality planners develop control scenarios with reduced emissions of air pollutant species of interest from specific emissions sources. Multiple scenarios are then modelled to
determine which control strategies have the desired result of bringing air pollutant concentrations below some threshold or standard.

Emission inputs relevant for regulatory modeling are generated from the bottom up using a wealth of data describing the emission factors and activity characteristics of thousands of sources. The preparation of an emissions inventory, which seeks to describe the annual emissions of every relevant process, is a complex multi-year effort. Further, the spatial (both horizontal
and vertical), temporal (seasonal, weekly, hourly, etc), and chemical speciation variability among these sources must be individually described and projected in order to be useful to the CTM system. Alternative emissions scenarios are generally not reconstructed afresh, but instead are modelled as variations from some base-case emissions scenario. Nonetheless, the preparation of alternative emissions scenarios is often a time-consuming step for air quality modeling applications, and repeated preparation of such inputs provides many opportunities for inconsistencies and user errors.

Additionally, air pollution research studies are often designed to characterize the fate and transport of novel pollutants, evaluate emerging chemical mechanism configurations or quantify the impact of updates to emissions speciation profiles (Qin et al., 2020; Lu et al., 2020). These kinds of detailed studies either do not warrant or cannot afford the effort required to generate entirely new bottom-up emission datasets, and the procedures required to introduce emissions of new species to existing input files are available, but again, expensive and error-prone. In response to these and other motivations, modules have been
developed for other modeling systems to process emissions inventories with activity data and chemical speciation within the CTM simulation. For example, Jähn et al. (2020) have added an online module to the climate and air quality model COSMO (Consortium for Small-scale Modeling) as well as an equivalent offline Python-based processing tool.

In older versions of the Community Multiscale Air Quality (CMAQ) model (version 5.2 and earlier), it is possible to adjust the emissions of a given species by a scaling factor that is applied across all emission sources, without having to modify the
underlying emissions files. However, there is no straightforward way to target modifications to a specific emissions sector or a geographic location, nor is there a way to modify the particle size distributions of emissions. Moreover, once a scaling operation has been applied, it is cumbersome for the user to directly determine that the operation proceeded correctly.



The Detailed Emissions Scaling, Isolation, and Diagnostic (DESID) module is designed to address these limitations. With DESID, which is a CMAQ-based module, it is possible to read an arbitrary number of gridded emissions files as well as point source emissions files, each representing a particular source sector or category (hereafter, an emissions *stream*). The modeler can then apply different scaling rules to adjust emissions from each stream, providing greater flexibility and precision in designing emissions sensitivity studies and exploring state-of-the-art chemical mechanism configurations. In addition, extensive details are written to the log file, and the option exists to create diagnostic files so that the user can be certain that the emissions have been adjusted as intended. Here we describe the concepts and implementation of DESID and provide several use cases that demonstrate its features. Though DESID was first included with CMAQ version 5.3 (Appel et al. 2020), a few refinements have been made subsequently. This paper describes the version as it exists in CMAQ version 5.3.2 (USEPA, 2020a). We conclude with some thoughts on potential future directions in emissions modeling for air quality applications.

## 2 Methods

### 2.1 Algorithm Framework

For standard and emerging applications, CMAQv5.3.2 relies on several offline gridded (e.g. area sources, motor vehicles, residential wood burning, volatile chemical products), offline point (e.g. wild- and prescribed fires, energy generation, industrial facilities, commercial marine), and online (e.g. biogenic vapors, wind-blown dust, sea-spray aerosol, lightning-generated nitric oxide) emission input streams (Fig. 1). CMAQ processes each of these types of streams differently (USEPA, 2020b): offline gridded emission rates are read in directly as arrays aligned with the model grid; offline point emission rates are read in, assigned to the appropriate horizontal row and column, and allocated vertically using source parameters (e.g. stack height, exit velocity, temperature) via buoyancy calculations; online emissions modules incorporate meteorological (e.g. sunlight, air temperature, relative humidity, wind speed) and geographical (e.g. land cover classification, leaf area index, ocean fraction) inputs to calculate emission rates. The offline emission inputs, in practice, already include significant chemical speciation while the online emission modules must speciate emission rates directly. Despite the differences among these three broad categories, DESID structures the flow of emissions processing within CMAQ so that each emission stream is retrieved, modified and diagnosed consistently and then incorporated independently.

Previous versions of CMAQ (5.2.1 and before) and many other CTMs employ processing approaches that vary among emissions streams. There are several reasons for these inconsistencies. Models become more complex over time due to increased computational capabilities and the evolving understanding of air pollution sources over multiple decades. In addition, there is a general lack of resources available for model refactoring and infrastructure development. For example, as Fig. 2 illustrates, CMAQv5.2.1 read emissions of gas-phase pollutants from offline gridded emissions, proceeded through several online and offline point streams, and then finally read all aerosol emissions at the same time. Because gas and aerosol emissions from the same streams were read and incorporated separately, transparent online scaling was not possible.





Additionally, CMAQ could read at most one offline gridded emission input file and several offline point input files. Because
of this limitation, sector-specific gridded streams were merged prior to input in CMAQ. Thus, all sector-specific information
was lost prior to inclusion in the CMAQ model and individual sectors required modification and reprocessing of upstream
files.

To overcome these and other limitations, DESID makes use of a series of generalized subroutines developed to handle critical
processing steps like emission rate retrieval, error checking (e.g. for negative values), size distribution allocation, and unit
conversions of all emissions streams (Fig. 3). With this uniform approach in place, model users and developers can be confident
that emissions are treated as expected across all streams. Several features that accommodate common emissions processing
tasks and alleviate workflow bottlenecks for research and regulatory applications build upon this robust system.

This section demonstrates the most useful features that have been incorporated into DESID to date. We begin with an
explanation of the Emissions Control Interface, a Fortran namelist file that specifies all rules and definitions for DESID
behavior. We specifically address how to add or perturb emissions of chemical species from any emission stream, incorporate
spatial dependence, expand scaling to multiple species or streams, ensure mass or mole conservation, and prescribe aerosol
size distributions. At this time, DESID does not support rules that vary in time (e.g. application of custom diel temporal profile),
but this feature is planned for a future release. Finally, we introduce the various features available for documenting the data
received by CMAQ from each emissions stream and the operations executed by DESID.

## 2.2 Working with DESID

### 2.2.1 Interface

The Emission Control Interface (ECI) provides a flexible and human readable platform for directing the behavior of DESID.
It is designed to accommodate typical CMAQ simulation configurations, basic perturbation cases and highly complex scaling
or mapping changes with minimal input lines arranged in a clear and concise layout. It manages these tasks while referencing
several environment variables defined in the CMAQ run script including aliases for offline emission streams. The publicly
available code for CMAQv5.3 and beyond contains default versions of the ECI to support emissions mapping for every
supported chemical mechanism including the following: Carbon Bond 6; the SAPRC07 (Statewide Air Pollution Research
Center) mechanism, and RACM2 (Regional Atmospheric Chemistry Mechanism). Without the ECI, CMAQ will assume
emissions are zero for all chemical species.

The ECI comprises four components to support its breadth of features: *Emissions Scaling*, *Region Definitions*, *Family
Definitions*, and *Aerosol Size Distribution Definitions* (Fig. 4 and Section S1). The *Emissions Scaling* component includes all
high-level rules to be executed, while the remaining three components provide definitions for more specific scaling choices.
First, we demonstrate common scaling rules possible in the *Emissions Scaling* component that do not require additional
definitions from the support components. The *Region*, *Family*, and *Aerosol Size Distribution* definitions are described
subsequently. Additional details and tutorials can be found in the CMAQ user guide (see Code and Data Availability section).



### 2.2.2 Emissions Scaling

The *Emissions Scaling* component is formatted as a table with a user-defined number of rows, each corresponding to an individual rule. These rules may be logistically simple (e.g. map the variable named NO from one emissions stream directly to the CMAQ species NO) or considerably complex (e.g. scale multiple species by 75% that have already been mapped for all
emission streams). During the CMAQ initialization process, DESID reads these rules and translates them into a series of low-level instructions to be carried out during each model time step. Each instruction involves at most one emission stream, one CMAQ variable and one emission variable. The simplest rules usually translate to one instruction, but the more complex ones (e.g. affecting multiple streams or multiple CMAQ species) are made up of several instructions.

Each rule is articulated with 8 fields (Table 1). Examples 1 and 2 in Table 2 demonstrate rules to map NO and fine-mode
elemental carbon (EC), respectively, for all emission streams. More complicated scaling rules are possible by exercising options in the available fields. For these examples, the add ('a') operator is used in the 'Op' field, indicating that these emissions should be added to the system. Example 3 shows how to modify the instructions created by Example 1 so that NO emissions are multiplied by 0.8 using the multiply 'm' operator. Example 4 achieves the same result but using the overwrite operator 'o'. Because DESID processes rules in the order they are provided in the ECI, if the 'm' operator were used in Example 4, the
scale factor applied to all NO emissions would equal 0.64 (0.8 * 0.8). The 'a' operator may be used for a rule with the same emissions and CMAQ variables to add or subtract emissions by using positive or negative scale factors. For example, if Example 1 appeared twice in the ECI, then 200% of the NO emissions would be incorporated into the CMAQ simulation.

Examples 5 and 6 demonstrate the approach to add multiple emission variables together to contribute to one CMAQ variable. In this case, particulate organic carbon (POC) and particulate non-carbon organic matter (PNCOM) are combined to contribute
to the emissions of aerosol primary organic matter (APOM). If one wants to scale the CMAQ variable, APOM in this case, it would usually make sense to scale the rates of both contributing emissions variables, POC and PNCOM. This can be achieved using Example 7, which multiplies the emissions of APOM for both emission variables by 150%. Example 8 demonstrates how subtraction may be used to return the emissions of APOM to its original 1:1 mapping with the emission streams.

The examples so far have assumed that the units of the emissions and CMAQ variables are equivalent. However, gas emissions
in CMAQ are specified in molar units (mol s$^{-1}$) while aerosols are in mass units (g s$^{-1}$) per grid-cell. In order to map CMAQ species to emissions variables with differing units as in Example 9, the BASIS field is helpful for prescribing unit conversions. Example 9 has dictated that mass be conserved when scaling additional emissions of APOM from gas-phase carbon monoxide (CO) from all streams. In this case, DESID will first convert the emission rate of CO to mass units using the molecular weight of CO, and then multiply by the scale factor (2%). If MOLE were chosen instead, DESID would (in this case) first scale CO
emissions to 2% (since gas phase species are typically provided by mole) and then convert to mass units (since aerosol species are tracked in CMAQ in terms of mass) using the molecular weight of APOM. In Examples 1-8 and 10, no unit conversions are performed. To perform mass and mole conserving calculations DESID must know the molecular weight of any emission





variable to be converted. A table is provided within the CMAQ source code (EMIS_VARS.F) that stores the molecular weights of all the known emission variables produced by the Sparse Matrix Operator Kernel Emissions (SMOKE) modeling system.

New emission variables and corresponding molecular weights should be added to this table as needed.

Finally, another common need for emission sensitivity cases is to target one emission stream for mapping or scaling modification. Example 10 demonstrates a rule for overwriting the scale factor for NO emissions to 200%, but only for the offline stream labelled ONROAD. Labels for offline gridded and point streams are set in the CMAQ run script using environment variables of the format GR_EMIS_LAB_xxx and STK_EMIS_LAB_yyy, respectively. The xxx and yyy indices

correspond to the environment variables that store the filenames of each offline stream, GR_EMIS_xxx and STK_EMIS_yyy, respectively. Online streams have default stream labels that may be used for scaling rules (see Table 1). An important consideration when designing an emissions input dataset is the level of source detail—DESID can only modify emissions for a specific source if that source is provided as its own stream.

### 2.2.3 Region Definitions

The *Region Definitions* component of the ECI maps labels for gridded spatial arrays to the input files and variables containing data for those arrays. Each entry or row in this component contains three fields (Table 3), which identify the input file and target variable to be associated with a specific region. The data for each region is expected to align with the simulation domain resolution and projection and include real numbers between 0 and 1.0, quantifying the fraction of emissions in each model grid cell that is associated with the region. Common examples of regions used for scaling include political areas like countries,

states, or counties or geographical features like oceans, lakes, or forests. Data files containing variables describing political boundaries are available for a typical 12 km continental U.S. domain from the Community Modeling and Analysis System Data Warehouse (USEPA, 2019a). Tutorials demonstrating a process for creating custom region variables for any grid using open-source tools will be available in future CMAQ repositories. As described in Table 3, if all the variables in an input file are desired (e.g. all of the lower 48 U.S. States), then the ALL keyword may be used for the REGION LABEL and TARGET

VARIABLE to instruct DESID to make all of the variables on the input file available as regions, reducing the number of input lines from 48 to 1. There is no limit to the number of input files that may be referenced and read to define regions in DESID.

Table 4 demonstrates three examples defining regions of increasing size—a U.S. city, Chicago, a U.S. state, Illinois, and a U.S. regional expanse, the OHIO_VALLEY, are all defined. A hypothetical use case for prescribing NO₂ emissions using these regions is shown in Table 5. Example 11 maps the NO2 emissions variable to the CMAQ NO2 variable for the entire

domain. Example 12 articulates a sensitivity whereby emissions in the Ohio Valley are cut by 30%. Examples 13 and 14 refine this perturbation with further spatial detail, overwriting the 30% cut with a 10% increase and an 80% decrease in emissions for the state of Illinois and Chicago area, respectively. The *Region Definition* feature thus facilitates implementation of highly refined spatially dependent emissions sensitivity experiments within CMAQ. However, these modifications are currently only





possible at the resolution of the simulation grid. The ECI for enforcing example 13 and the resulting fields of emission and

$NO_2$ concentration changes are given in supporting information (Section S2 and Fig. S1).

### 2.2.4 Family Definitions

Emissions sensitivity experiments can of course require perturbation of more than one chemical species, emission stream, or region simultaneously, and often these perturbations are articulated with the same relative increase or decrease to all species, streams, or regions involved. This kind of across-the-board forcing can be representative of changes in technology or a change

in the market share of pollutant sources. With the examples shown so far, highly detailed emissions perturbations are possible, but in order to apply them to many species, for example, repetition is the only option. To alleviate this inefficiency, the *Family Definition* component provides an interface for populating groups of chemical species, emission streams, and regions, so they may enhance the impact of each scaling rule. Table 6 gives an example of each type of family possible with DESID. The chemical family example creates a group of aromatic species names AROMATICS. The example stream family, INDUS,

groups emissions from industrial sources including power generation. The region family combines several states in the southwest U.S. into a group labeled SOUTHWEST.

Several examples using these groups appear in Table 7. Before AROMATICS can be used in Example 19, its members must be mapped individually to emissions variables (Examples 15-18). Example 20 shows how the scale factor for NO emissions in 5 states can be overwritten simultaneously, and Example 21 combines the functionalities for chemical and stream families

to overwrite the emissions of all four aromatic compounds from the group of industrial sources defined in Table 6. An ECI enforcing these examples and the resulting emissions concentration fields are given in the supporting information (Section S3 and Fig. S2-S3). These simplifying features greatly shorten the repetition required in the ECI and enhance its utility.

### 2.2.5 Aerosol Size Distribution Definitions

The details of aerosol size distributions are often overlooked when applying CTMs because most particulate matter

performance evaluations and model predictions are presented in terms of bulk $PM_{2.5}$ or $PM_{10}$ mass. Meanwhile, particle size is a critical parameter for model processes like condensational growth, heterogeneous reactions, dry deposition, and wet scavenging, each of which have important impacts on the burden of PM and gaseous pollutants of concern. The potential significance of ultrafine particles for human health impacts and climate-scale feedbacks also continues to grow (USEPA, 2019), especially in large population centers and near emission sources. An important aspect of predicting atmospheric particle

sizes is applying realistic size distributions to primary particle emission rates. Although data are relatively sparse, several studies have collected particle size estimates to represent broad sectors of emissions (Winijkul et al., 2015; Boutzis et al., 2020). Although these datasets are valuable and should be used to further develop existing emission inventories, there are considerable uncertainties with applying size distributions uniformly across all members of a sector. DESID, therefore,





supports online application of primary particle size distributions to facilitate both research and quality control of this aspect of
emissions modeling.

The *Aerosol Size Distribution Definition* component maps individual emission streams to size distributions available in a table
compiled with the CMAQ source code (Table 8). This table, called em_aero_ref and found in AERO_DATA.F, defines the
parameters needed to distribute the mass of emissions to particle size categories (Table 9). These parameters include the mass
fraction present in each aerosol mode, the mode geometric mean diameter, and the standard deviation describing each mode's
assumed log-normal distribution (Binkowski and Roselle, 2003). Because emissions inventories (i.e. the U.S. National
Emission Inventory) generally distinguish fine and coarse PM, it is recommended that separate rows be included to process
fine and coarse species. By default, DESID maps the FINE and COARSE distribution labels of all emission streams to the
FINE_REF and COARSE_REF size distributions documented by Nolte et al. (2015). They may be overridden by subsequent
entries though, as shown for the AIRCRAFT stream in Table 8. In this example, specific distribution labels are assigned for
the WILDFIRE stream rather than using the existing labels FINE and COARSE.

With the *Aerosol Size Distribution Definition* component populated and size distribution parameters available for each stream,
scaling rules can be applied with those distribution labels in the PHASE/MODE field (Table 10). In Examples 22 and 23, a
fraction of fine- and coarse-mode particulate nitrate are mapped to the emissions for fine-mode nitrate (PNO3) and coarse-
mode PM (PMC), respectively. For the coarse-mode nitrate, a scale factor of 0.048% quantifies its mass contribution of PMC
emissions from all streams. As DESID processes this rule, it will reference the stream-specific size distributions mapped to
FINE and COARSE and assign mass to the appropriate aerosol size modes defined internally (e.g. Aitken-, accumulation- and
coarse-mode nitrate; ANO3I, ANO3J and ANO3K, respectively). Examples 24 and 25 show how the size distribution for
particulate nitrate can be reassigned to distributions specific for wildfire emissions.

### 2.3 Diagnostics

DESID provides a variety of features of varying complexity to support the vast majority of emissions sensitivity scenarios that
air quality modelers would find useful. As this complexity grows, however, quality assurance becomes a crucial consideration.
Thus, the new emissions module includes three important types of updates to protect against mistakes and instill confidence
in results. First, DESID incorporates error-checking for all user inputs to catch trivial inconsistencies (e.g., typographical errors
or missing data fields). In addition, if scaling rules reference an emission variable, stream, region, or CMAQ variable that is
not available, DESID will abort, unless users override this behavior. Second, the module outputs relevant messages to the
human readable CMAQ log files to confirm processing of scaling rules and other emission inputs. Users should examine this
log file to confirm that there are no unintentionally unused emissions variables, that families are defined as intended, that
stream-specific size distributions are mapped correctly, and that regions are mapped correctly. An exhaustive list is then printed
containing the scale factors applied to the emissions of every CMAQ species from every stream so that users can see directly
how a set of scaling rules were interpreted by DESID. Finally, DESID optionally outputs gridded data files with the mapped,





scaled and processed emissions for each stream, including particle number and surface area emissions, which are calculated online using the stream-specific size distribution parameters and mass emission rates. There are three formats available for outputting these data including surface-layer only, full three-dimensional field, and two-dimensional column sum. Not only can these outputs then be used to confirm correct scaling, but they can also be used to compare emissions from offline gridded, offline point and online sources on a consistent data grid and used as inputs for subsequent simulations.

## 3. Relevant Applications

The development of DESID features was catalyzed by recognized needs in the air quality modeling community. Emission perturbation studies are a fundamental aspect of air quality modeling and analysis and include important objectives like source attribution, estimation of the benefits of policies under consideration, and trends analysis. For example, in the 2012 PM National Ambient Air Quality Standard Regulatory Impact Analysis (USEPA, 2012), multiple annual emission fields were developed that reduced emissions of specific $PM_{2.5}$ precursors by fixed percentages in selected regions to inform modeling of the emission reductions needed to meet standard levels. The modification of a base emissions dataset to develop many new emission datasets in such applications is costly, time-consuming, and requires storage. With DESID, these relatively straightforward perturbation cases can be directly programmed, executed, and confirmed with zero increased storage cost (unless diagnostic files are written). Moreover, this aspect is of high value for deployment of CTMs on cloud-computing platforms where storage needs are monetized and producing alternative emission input files can result in significant additional costs.

Beyond introducing efficiencies for standard perturbation exercises, DESID benefits have also been demonstrated for air quality research efforts, specifically for improving speciation of bulk pollutants like PM and volatile organic compounds (VOC). For example, organic PM mass and ozone predictions are significantly impacted by emissions of primary organic aerosol (POA) and emissions of volatile chemical products (VCPs) (Lu et al., 2020; Qin et al., 2020). For more than a decade, POA emissions have been demonstrated to partition dynamically between the particle and gas phases (Robinson et al., 2007). To account for this behavior, the POA emission rate is typically distributed from one emission variable to several CTM species, each with a different volatility. These volatility distributions vary among source types. Some sources, like motor vehicles, are relatively well-understood, while others, like biomass burning, are exceedingly complex and remain challenging despite increasing attention. VCPs have recently received increased attention as it has been acknowledged that their role as sources of carbon pollution has increased as other sources (e.g. vehicles, industry) have become cleaner through regulatory actions. Although VCPs have been treated by emission inventories for decades, there are large uncertainties in their estimation and speciation methods that are currently being addressed. Over time, the data gathered from the research community for POA and VCPs must be incorporated into existing operational emissions inventory and modeling tools. Part of that evolution though, involves using CTMs to reduce the uncertainty in the updated parameters, quantify changes in PM model performance, and estimate the impact they have on strategies for attaining ambient air quality standards. For example, proposed VCP speciated



emissions can be scaled online to typical reference pollutants like CO or non-methane organic gases (NMOG). To summarize, DESID features allow researchers to bypass creation of alternative bottom-up emission datasets or extensive modification of

input files leading to greater transparency, automated documentation of experimental scale factors, and more time for data interpretation.

These features are further useful for integrating emissions data from multiple inventories and modeling methods, which may be an asset for state-of-the-art regional- and global-scale chemical transport modeling. Matthias et al. (2018) reviewed the landscape of top-down and bottom-up approaches for creating inventories and applying spatiotemporal allocation to generate

emissions for air quality models throughout the world. While noting the benefits of integrating emerging big data sources (e.g. traffic data, agriculture practices) into strategies for creating emission inputs, they also stressed that inclusion of more data can sometimes introduce high uncertainties as well as discontinuities along, for example, political boundaries. By allowing users to employ any number of emissions files as independent data streams and apply region-based scaling to activate or deactivate particular streams in specific areas of the modeling domain, DESID makes it feasible to explore hybrid configurations of

emission inputs from a variety of datasets.

Finally, the standardization of inputs via the ECI makes possible the automation of emission perturbation cases, which is useful for several key applications, including coupling with energy system models and generating input datasets for reduced-form models. Energy system optimization models such as the MARKet ALlocation (MARKAL) model facilitate the development of scenarios that project the evolution of the energy system and its associated emissions decades into the future under differing

assumptions about energy demands and the costs and availability of technologies and fuels. In previous efforts to link energy system projections to emissions and CTMs (Loughlin et al. 2011; Gonzalez-Abraham et al. 2015; Ran et al. 2015), regional and sectoral growth factors from MARKAL were applied to the relevant intermediate files from a base-year inventory, and the modified sectors were then remerged prior to running CMAQ. Using DESID, the workflow becomes far simpler, with region- and stream-specific growth factors from the energy system model directly incorporated into the ECI.

To facilitate the optimization of emission control strategies over many possible cases (Huang et al., 2020; Fu et al., 2006; Cohan et al., 2006), response-surface models (RSMs) have been developed by fitting statistical models to the output of many CMAQ simulations (Xing et al., 2011, 2017). Although deep learning methods may reduce the computational burden of RSM development (Xing et al., 2020), dozens of CMAQ simulations are still needed to sample the emission control space in developing RSMs for typical air quality management applications. DESID greatly simplifies the implementation of the CMAQ

simulations for RSM development by eliminating the need to create dozens of sets of emission input files.  Further, the latest version of the RSM-VAT (Response Surface Model – Visualization and Analysis Tool) software developed as part of the Air Benefit and Cost Attainment Assessment System (ABaCAS; http://www.abacas-dss.com) includes a module to auto-generate ECIs for the suite of CMAQ simulations needed in RSM fitting. DESID features thus benefit the wide range of regulatory and research applications of the ABaCAS and broader air quality modeling communities.





## 4. Conclusions and Future Directions

Bulk emission rates and chemical composition persist as a major source of uncertainty impacting air quality model performance and predictions. Therefore, it is important to make algorithms available that reduce the logistical burden of exploring these uncertainties. In this way, the research community can build greater confidence in its understanding of atmospheric science fundamentals, the policy community can build greater confidence in the likelihood of success of policy scenarios simulated by these CTMs, and the regulatory community can better understand the contribution of individual sources to important atmospheric pollutants.

Conventional approaches for manipulating emission inputs offline and typically require a sequence of several steps that often render them time consuming and error prone. Over the past 15 years, CMAQ has gradually evolved in the direction of doing more of its emissions calculations online. Sea spray emissions initially existed only in the coarse mode and were chemically inert in CMAQ v4.3 (released in 2003). With CMAQ v4.5 and the AERO4 module, sea spray emissions were computed online as a function of meteorology using an OCEAN file specifying the fraction of each grid cell that is open ocean or surf zone (Zhang et al. 2005; Appel et al. 2008). Other than sea spray, all emissions were calculated offline in SMOKE, and CMAQ read in a single, large, 3-D emissions input file. The capability to read point source emissions and calculate their plume rise online, as well as the ability to calculate biogenic emissions online, were both added in CMAQ v4.7 (Foley et al., 2010). Bidirectional flux of mercury (Bash 2010) and ammonia (Pleim et al. 2013) and lightning-generated emissions of $NO_x$ (Allen et al. 2012) became available in CMAQ v5.0. Marine halogen emissions were added to represent iodine and bromine chemistry (Sarwar et al., 2019) in CMAQ v5.2. DESID achieves an important step towards further unifying emissions and atmospheric chemistry and transport into a holistic modeling framework.

By supporting emission rate manipulations across a range of complexity online in CMAQ, DESID enhances transparency, automates documentation, reduces the number of trivial errors, and ultimately saves resources. DESID's Emission Control Interface allows users to enforce simple mapping and scaling rules or configure broadly defined sensitivity scenarios that modify multiple chemical species and/or emission streams, potentially over one or several spatial regions of interest. For the first time, users also have stream-specific control over the aerosol size distributions assumed for each emission source. Importantly, DESID standardizes inputs and definitions of variables, thereby reducing the required level of expertise required to use CMAQ's internal algorithms. The module accomplishes this with minimal increase in computational burden. For example, a simulation with source-specific aerosol size distributions and diagnostic output applied for 27 and 19 offline gridded and point emission files, respectively, increased model run-time by an average of 3.5% for 10 summertime simulation days compared to a reference case with 2 and 8 offline gridded and point files, one primary aerosol size distribution, and no diagnostic output. As the science in CMAQ evolves (e.g. chemical mechanisms, aerosol microphysics configurations), users can have confidence that DESID will coevolve with it, thus removing the burden to update offline approaches. The features available in DESID support a broad range of applications from routine regulatory-oriented perturbation cases, to atmospheric chemistry research efforts and coupling with external models (e.g. energy system models, reduced-form models).

Future developments in DESID will further support air quality policy and research analysis by incorporating other common offline tasks. These include interpolating gridded emissions to the selected model projection and domain, reassigning the diel
profile of emissions from specific sources, and allowing creation of experimental point and area sources online using the ECI. This latter feature will be particularly important for modern air quality issues like quantifying impacts from forest fire plumes and characterizing the regional burden of pollutants of immediate concern like per- and polyfluoroalkyl substances and ethylene oxide releases.

**Code and Data Availability**

CMAQ source code, including Emission Control Interfaces for every supported chemical mechanism, is freely available via https://github.com/usepa/cmaq.git. Archived CMAQ versions are available from the same repository. Although DESID is available in version 5.3 and later, the most recent version 5.3.2 is the default recommendation. Model input data are available from the Community Modeling and Analysis System (CMAS) Data Warehouse (https://doi.org/10.15139/S3/MHNUNE). Additional details regarding DESID formulation and its relationship to other CMAQ modules are given in the CMAQ user
guide, Appendix B (https://github.com/USEPA/CMAQ/tree/master/DOCS/Users_Guide) and a comprehensive tutorial is available at
(https://github.com/USEPA/CMAQ/blob/master/DOCS/Users_Guide/Tutorials/CMAQ_UG_tutorial_emissions.md).

**Supplement**

The supplement related to this article is available online at:

**Competing Interests**

The authors declare that they have no conflict of interest.

**Disclaimer**

The views expressed in this article are those of the authors and do not necessarily represent the views or policies of the U.S. Environmental Protection Agency.



**Acknowledgements**

This work was funded by the U.S. EPA Air and Energy Program. The authors would like to thank Kirk Baker, Kristen Foley, Barron Henderson, Christian Hogrefe, William Hutzell, Shawn Roselle, and Golam Sarwar for valuable feedback, testing and application during the creation and integration of DESID.

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



**Table 1. Fields required for articulating emission scaling rules in DESID.**

| Field Name | Description |
|---|---|
| Region Label | Identifies the region over which the rule is to be applied. The keyword EVERYWHERE applies the rule to all grid cells. |
| Stream Label | Identifies the emission streams for which the rule is to be applied. These labels are defined for offline gridded and point sources in the CMAQ runscript. Online emission streams use keywords including BIOG (biogenic vapors), MGEM (marine gas), LTNG (lightning NO), WBDUST (wind-blown dust), and SEASPRAY (sea spray aerosol). The keyword ALL applies the rule to all streams, including online streams. |
| Emission Variable | Identifies the variable from the emission stream file or online module that emissions should be mapped from. The keyword ALL may be used to apply the rule to all previously mapped variables. |
| CMAQ Species | Identifies the variable within CMAQ that the rule should be applied to. The keyword ALL may be used to apply the rule to all previously mapped Species. |
| Phase/Mode | Used to distinguish gas or aerosol calculations. If the CMAQ variable is a gas, this field should be set to GAS. If it is an aerosol, this field should indicate the desired aerosol mode (e.g. COARSE, FINE, or other user-defined options) or use the keyword AERO to apply the rule to the entire distribution. |
| Scale Factor | Real-valued number applied for scaling calculations. |
| Basis | Determines whether mass or moles are conserved during conversion from gas to particle emission rates or vice-versa. The keyword MASS conserves mass, MOLE conserves moles, and UNIT performs no conversions. |
| Op | Determines the operation for each rule to apply: 'a' adds the rule to the existing instruction set, 'm' finds existing instructions matching this rule's features (i.e. variable names, stream labels, etc) and multiplies their existing scale factors by this rule's scale factor, 'o' finds existing instructions matching this rule's features and overwrites their scale factors with this rule's scale factor. |





**Table 2. Examples of basic scaling rules in the Emissions Scaling component.**

| Example | Region Label | Stream Label | Emission Variable | CMAQ Species | Phase/Mode | Scale Factor | Basis | Op |
|---|---|---|---|---|---|---|---|---|
| 1 | EVERYWHERE | ALL | NO | NO | GAS | 1.0 | UNIT | a |
| 2 | EVERYWHERE | ALL | PEC | AEC | FINE | 1.0 | UNIT | a |
| 3 | EVERYWHERE | ALL | NO | NO | GAS | 0.8 | UNIT | m |
| 4 | EVERYWHERE | ALL | NO | NO | GAS | 0.8 | UNIT | o |
| 5 | EVERYWHERE | ALL | POC | APOM | FINE | 1.0 | UNIT | a |
| 6 | EVERYWHERE | ALL | PNCOM | APOM | FINE | 1.0 | UNIT | a |
| 7 | EVERYWHERE | ALL | ALL | APOM | FINE | 1.5 | UNIT | m |
| 8 | EVERYWHERE | ALL | ALL | APOM | FINE | -0.5 | UNIT | a |
| 9 | EVERYWHERE | ALL | CO | APOM | FINE | 0.02 | MASS | a |
| 10 | EVERYWHERE | ONROAD | NO | NO | GAS | 2.0 | UNIT | o |






**Table 3. Fields required for articulating region scaling rules in DESID.**

| Field Name | Description |
| --- | --- |
| Region Label | Label for the region being defined. The ALL keyword may be used (in conjuction with setting TARGET VARIABLE to ALL) to pass through the data and names of every variable on the input file. |
| File Label | Identifies the file containing data for this horizontal region. The FILE LABEL should be equivalent to the name of an environment variable defined in the CMAQ runscript storing the file path and name. Any number of input files may be referenced during a simulation. |
| Target Variable | Identifies the variable from the input file to be associated with this REGION LABEL. The ALL keyword may be used (in conjuction with setting REGION LABEL to ALL) to pass through the data and names of every variable on the input file. |





**Table 4. Example entries for Region Definition component.**

| Region Label | File Label | Target Variable |
| --- | --- | --- |
| CHICAGO | US_CITIES | CHI |
| IL | US_STATES | ILLINOIS |
| OHIO_VALLEY | US_REGIONS | OHIO_VALLEY |



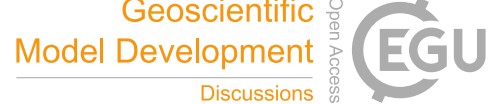

**Table 5. Example scaling rules using region labels.**

| Example | Region Label | Stream Label | Emission Variable | CMAQ Species | Phase/Mode | Scale Factor | Basis | Op |
|---------|-------------|--------------|-------------------|--------------|------------|--------------|-------|-----|
| 11 | EVERYWHERE | ALL | NO2 | NO2 | GAS | 1.0 | UNIT | a |
| 12 | OHIO_VALLEY | ALL | NO2 | NO2 | GAS | 0.7 | UNIT | m |
| 13 | IL | ALL | NO2 | NO2 | GAS | 1.1 | UNIT | o |
| 14 | CHICAGO | ALL | NO2 | NO2 | GAS | 0.2 | UNIT | o |





**Table 6. Examples of families supported by DESID.**

| Family Type | Family Name | Members |
|---|---|---|
| Chemical | AROMATICS | TOL, XYLMN, BENZENE, NAPH |
| Stream | INDUS | POINT_EGU, POINT_NONEGU, POINT_OTHER |
| Region | SOUTHWEST | CA, NM, AZ, NV, UT |






**Table 7. Scaling examples using chemical, stream, and region families.**

| Example | Region Label | Stream Label | Emission Variable | CMAQ Species | Phase/Mode | Scale Factor | Basis | Op |
|---|---|---|---|---|---|---|---|---|
| 15 | EVERYWHERE | ALL | TOL | TOL | GAS | 1.0 | UNIT | a |
| 16 | EVERYWHERE | ALL | XYLMN | XYLMN | GAS | 1.0 | UNIT | a |
| 17 | EVERYWHERE | ALL | BNZ | BENZENE | GAS | 1.0 | UNIT | a |
| 18 | EVERYWHERE | ALL | NAPH | NAPH | GAS | 1.0 | UNIT | a |
| 19 | EVERYWHERE | ALL | AROMATICS | AROMATICS | GAS | 0.6 | UNIT | m |
| 20 | SOUTHWEST | ALL | NO | NO | GAS | 0.9 | UNIT | o |
| 21 | EVERYWHERE | INDUS | AROMATICS | AROMATICS | GAS | 0.3 | UNIT | o |





**Table 8. Example of mapping in Aerosol Size Distribution Definition Component**

| Stream Label | Distribution Label | Distribution Reference |
|---|---|---|
| ALL[a] | FINE | FINE_REF |
| ALL[a] | COARSE | COARSE_REF |
| AIRCRAFT | FINE | AIR_FINE |
| AIRCRAFT | COARSE | AIR_COARSE |
| WILDFIRE | WILD_FINE | FIRE_FINE |
| WILDFIRE | WILD_COARSE | FIRE_COARSE |

[a] These entries are implemented in the DESID source code by default.

**Table 9. Examples of aerosol size distributions available within CMAQ**

| Distribution Reference | Weight Fraction | | | Diameter[a] | | | Standard Deviation | | |
|---|---|---|---|---|---|---|---|---|---|
| | Aitken | Acc | Coarse | Aitken | Acc | Coarse | Aitken | Acc | Coarse |
| FINE_REF | 0.1 | 0.9 | 0.0 | 60 | 280 | -[b] | 1.7 | 1.7 | - |
| COARSE_REF | 0.0 | 0.0 | 1.0 | - | - | 6000 | 1.7 | 1.7 | 2.2 |
| AIR_FINE | 1.0 | 0.0 | 0.0 | 30 | - | - | 1.3 | - | - |
| AIR_COARSE | 0.0 | 0.0 | 1.0 | - | - | 5000 | - | - | 2.1 |
| FIRE_FINE | 1.0 | 0.0 | 0.0 | 130 | - | - | 1.7 | - | - |
| FIRE_COARSE | 0.0 | 0.0 | 1.0 | - | - | 7000 | - | - | 1.8 |

[a] geometric mean diameter of the aerosol volume distribution
[b] parameters omitted for the purpose of this table when weight fraction is 0.0.


**Table 10. Examples of scaling rules using alternative size distributions for primary particulate nitrate emissions.**

| Example | Region Label | Stream Label | Emission Variable | CMAQ Species | Phase/Mode | Scale Factor | Basis | Op |
|---|---|---|---|---|---|---|---|---|
| 22 | EVERYWHERE | ALL | PNO3 | ANO3 | FINE | 1.0 | UNIT | a |
| 23 | EVERYWHERE | ALL | PMC | ANO3 | COARSE | 0.00048 | UNIT | a |
| 24 | EVERYWHERE | WILDFIRE | PNO3 | ANO3 | WILD_FINE | 1.0 | UNIT | o |
| 25 | EVERYWHERE | WILDFIRE | PMC | ANO3 | WILD_COARSE | 0.00048 | UNIT | o |






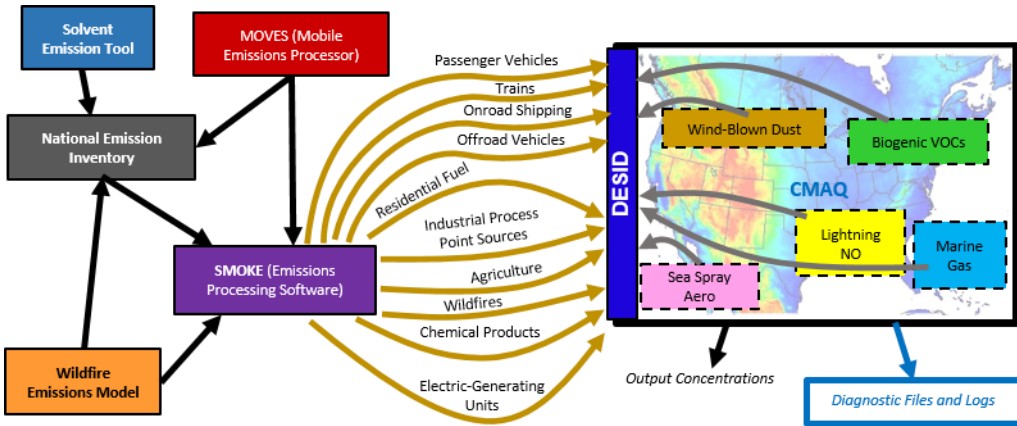

Figure 1. Example of potential emissions streams and data sources used that inform CMAQ. The offline streams and emission models depicted are not an exhaustive list of all data sources that contribute to a standard CMAQ simulation for the United States.



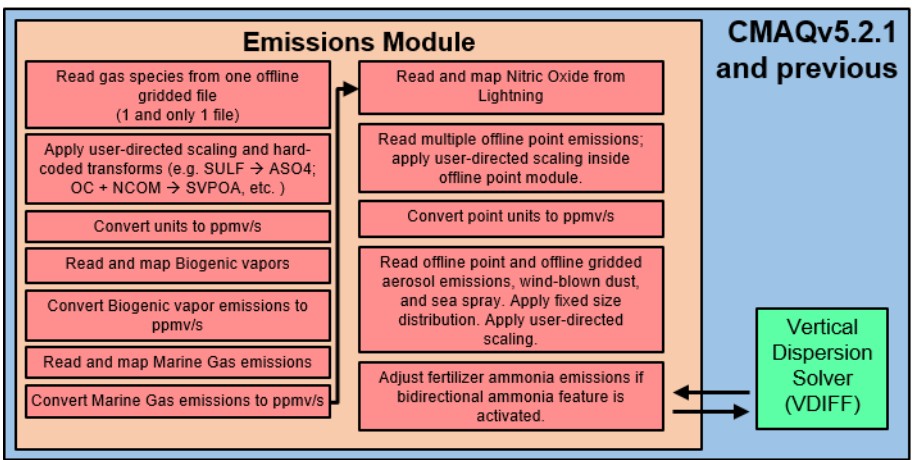

Figure 2. Algorithm for emission processing used by CMAQv5.2.1 and previous.

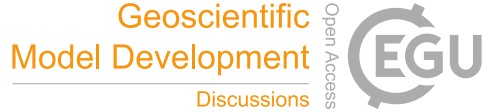

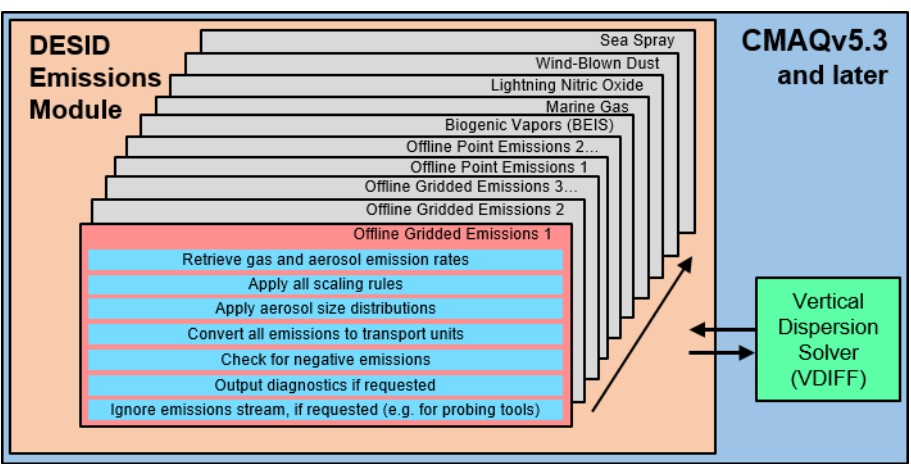

Figure 3. Algorithm for emission stream processing used by DESID. Emission rates are processed for each emission stream independently before DESID proceeds to the next stream.






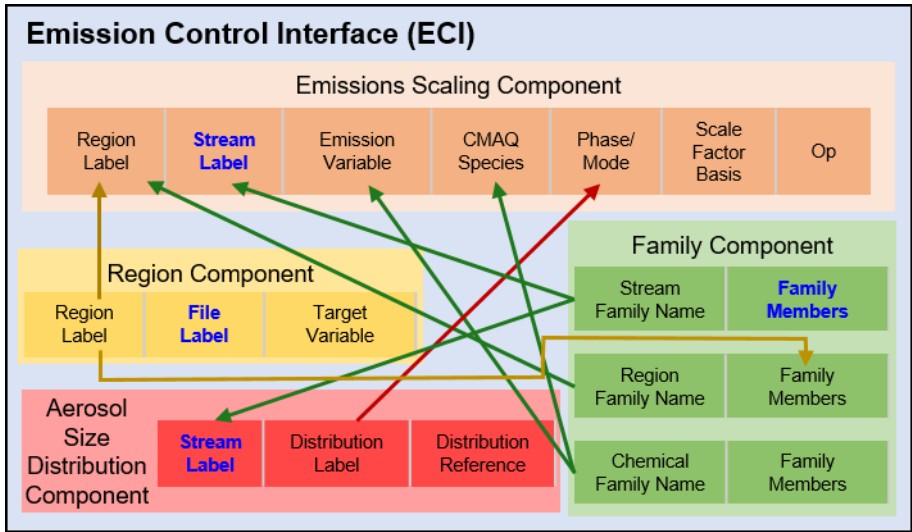

Figure 4. Schematic of Emission Control Interface (ECI) and flow of input options among individual components. Information flow lines are colored based on the component of origin. Text in blue indicates elements that should refer to environment variables set in the CMAQ run script except in the case where Stream Labels are populated from the Family Component with members that then refer to the run script. The "Distribution Reference" of the Aerosol Size Distribution Component refers to entries populated within the CMAQ source code.