# Peer review of "The Detailed Emissions Scaling, Isolation, and Diagnostic (DESID) module in the Community Multiscale Air Quality (CMAQ) Modeling System version 5.3.2"

_Geoscientific Model Development, 2020_

## Short Comment (SC1) · 21 Dec 2020

Dear authors,

in my role as Executive editor of GMD, I would like to bring to your attention our Editorial version 1.2:

https://www.geosci-model-dev.net/12/2215/2019/

This highlights some requirements of papers published in GMD, which is also available

on the GMD website in the 'Manuscript Types' section: http://www.geoscientific-model-development.net/submission/manuscript_types.html

In particular, please note that for your paper, the following requirement has not been met in the Discussions paper:

- Code must be published on a persistent public archive with a unique identifier for the exact model version described in the paper or uploaded to the supplement, unless this is impossible for reasons beyond the control of authors. All papers must include a section, at the end of the paper, entitled "Code availability". Here, either instructions for obtaining the code, or the reasons why the code is not available should be clearly stated. It is preferred for the code to be uploaded as a supplement or to be made available at a data repository with an associated DOI (digital object identifier) for the exact model version described in the paper. Alternatively, for established models, there may be an existing means of accessing the code through a particular system. In this case, there must exist a means of permanently accessing the precise model version described in the paper. In some cases, authors may prefer to put models on their own website, or to act as a point of contact for obtaining the code. Given the impermanence of websites and email addresses, this is not encouraged, and authors should consider improving the availability with a more permanent arrangement. Making code available through personal websites or via email contact to the authors is not sufficient. After the paper is accepted the model archive should be updated to include a link to the GMD paper.

As GitHub is not a persistent archive, please provide a persistent release for the exact source code version used for the publication in this paper. As explained in https://www.geoscientific-model-development.net/about/manuscript_types.html the preferred reference to this release is through the use of a DOI which then can be cited in the paper. For projects in GitHub a DOI for a released code version can easily be

created using Zenodo, see https://guides.github.com/activities/citable-code/ for details. Finally note, that according to our new Editorial (v1.2) all data and analysis / plotting scripts should be made available.

Yours, Astrid Kerkweg

———————————————

---

## Author Comment (AC1) · 21 Dec 2020

Dear Editorial Staff and Readers,

Thank you to the GMD Executive editor for pointing out the missing persistent DOI reference to the code we used for this manuscript. Please note that we have included it as an 'Asset' to the submission and readers/reviewers may find it there. We will add this DOI to the manuscript as part of the revision process before publication in GMD.

Best wishes, Benjamin N. Murphy

---

## Referee Comment (RC1) · Anonymous Referee #1 · 1 Feb 2021

This paper describes a new feature of tracking and modifying emission inputs in the CMAQ v5.3. It is delightful to read the well-organized manuscript, with clear explanation of the workflow and functionality of DESID and its implementation in the CMAQ model. The new development is a desired tool that will make it easier for researchers and regulatory model users who are not an emission expert to study sophisticated emission perturbation simulations to quantify source-receptor relationships. This manuscript provides a detailed description of the new feature that can benefit the CMAQ user community. Therefore, I recommend the publication of this manuscript, with some suggestions to further improve the usability of this tool.

1) While the description is comprehensive, it will be useful for the authors to provide some use cases with figures showing the emission changes or CMAQ output changes. 2) It should be pointed out the using some of the DESID features will require generating and retaining sector-level emission files and all more space to run the model, although most of these files are two dimensional. 3) Figure 1 shows the workflow of DESID in which both natural and anthropogenic source s 4) Future direction: All inline emission processes, including sea-salt, dust, biogenic and plume rise, are included in the VDIFF subroutine. As the addition of DESID, the emission module may be too complicated to be treated as a sub-subroutine. Has the team considered having emission as a separate subroutine in the science process file?
* * *

---

## Referee Comment (RC2) · Anonymous Referee #2 · 6 Feb 2021

**General Comments**

Polutant emissions are a key input of chemical transport models, and many applications of these models require emissions modifications. The preparation and perturbation of input emissions is thus an important task for the use of chemical transport models, but this task is usually performed using off-line emissions processing systems. The successful use of emissions processing systems such as SMOKE, however, requires a high level of expertise and experience and typically requires time-consuming, very

detailed work to be performed in which it is easy to make mistakes and hard to track and document the process and the results. In addition, many chemical transport models now also contain in-line calculation of natural emissions that are influenced by hourly meteorology; these in-line emissions are difficult both to track by the user and to modify without code changes.

This paper describes DESID, a new module for the CMAQ chemical transport model. DESID is in essence a versatile but powerful post-processor for off-line emissions processing systems as it enables further manipulation of the emissions fields produced by an emissions processing system. The DESID module has been designed with a very flexible, compact, yet powerful user interface. Its use can save considerable emissions preparation time and reduce computer storage requirements while providing control over different emissions streams, error checking, and useful output documentation as well as optional output of processed emissions files. The authors correctly note that resources are seldom available for "model refactoring and infrastructure development", but this new module is a welcome exception to this usual constraint and it constitutes a significant addition to the well-established CMAQ air quality modeling system. It will likely be adopted quickly and widely by the large CMAQ user community.

I found this to be a well-structured and well-written paper. I recommend its acceptance after minor revision. To this end I have made a number of specific comments and suggestions below that I believe will improve the final version and that I hope the authors will consider.

**Specific Comments**

1. It is mentioned on page 2 that emissions processing must address temporal allocation, but the time variation of input emissions files and how DESID treats this variation is only mentioned once in the paper, on page 5, line 132, and I am not sure this mention is totally correct. Input emissions are commonly specified for each hour of the day. If the model time step is less than one hour, will DESID still apply the instruction rules

during each model time step as stated. This may be true for online emissions streams, but is it also true for offline emissions streams? A few words about how the DESID emissions module fits into the CMAQ operator sequence would be helpful as well as a mention of the day-to-day variation of offline emissions streams (i.e., day-specific or day-of-week specific and monthly or seasonal variation).

2. It is also mentioned on page 2 that emissions processing must address chemical speciation, but in the discussion of the Emissions Scaling rules and Table 1 in Section 2.2.2, it is not made clear that one of the eight rule fields, "Emission Variable", is not an inventory emission variable but rather is a speciated emission variable output by the emissions processing system.

3. I found the use of examples in Section 2 very helpful for understanding how DESID works. One point that was not clear to me, however, appears in lines 139-141. If two rules apply to the same region, stream, and emission variable, would this result in the second rule modifying the scaling factor of the instruction corresponding to the first rule? That is, can multiple rules sometimes result in a smaller number of instructions?

4. Section 2.2 does not say what is the minimum number of rules required by the DESID ECI. Can it be zero (i.e., no rules), or if not could the minimum number of rules be given in a new table in the Supplement?

5. An important constraint on DESID noted on line 167 is the degree of disaggregation of the emissions streams input by the model. In lines 340-341 the authors note a traditional treatment of emissions with 2 offline gridded and 8 offline point emissions files vs. a much more detailed treatment with 27 offline gridded and 19 offline point emissions files. The authors might consider adding a new table to the Supplement that lists and compares these two sets of emissions streams (with corresponding environment variables?) as an example of a simulation that makes full use of the capabilities of DESID and what this higher level of inventory disaggregation looks like.

6. I am concerned that the use of Regional Definitions, while very intuitive and appealing, might not always give exactly the expected results when applied to multiple emissions streams. The issue is how to handle grid cells that straddle two or more regions. The authors state on lines 172-173 that the gridded definition of a region should quantify "the fraction of emissions in each model grid cell that is associated with the region", but I would expect that the value of the fraction of emissions for a border grid cell would frequently vary with the emissions stream. For example, consider the spatial distribution of ammonia emissions from on-road motor vehicles for a grid cell straddling the Indiana-Illinois state line, which will depend on road network geometery, vs. ammonia emissions from fertilizer application, which will depend on the distribution of farms. In effect I think applying a Regional Definition is equivalent to applying the same spatial surrogate to all emissions streams. I definitely am not saying that this limitation should preclude the use of Regional Definitions, but if the authors agree with this concern, then it would be worth noting in the manuscript that some uncertainty will inevitably be associated with the apportionment of emissions between neighboring regions due to this issue. This minor caveat is particularly important for policy applications, where different fractional reductions may be under consideration for different jurisdictions but the actual modeled reductions may be slightly different. Also, I expect this relative uncertainty would likely increase as the size of a region decreases (Rhode Island, say, or a smaller city).

**Technical Corrections/Suggestions**

p. 2, l. 32   Perhaps "... managers in designing programs to improve urban- ..."

p. 2, l. 43   It is not clear here whether "sources" refers to individual facilities or to different source sectors or source types.

p. 3, l. 71   It is noted here that the paper describes the DESID module as it exists in CMAQ version 5.3.2. Should the paper title reflect this version number?

[Figure]

p. 3, l.77   Perhaps "... and online gridded (e.g., biogenic and marine vapors, ..."

p. 4, l. 103   Perhaps "The rest of this section demonstrates ..."

p. 5, 2nd paragraph. Change the order of the third and fourth sentences?

p. 5, l. 149   Should it be "map emissions variables to CMAQ species with differing units"?

p. 6, l. 157   Just to comment that the table name is not given here but a similar description on l. 221 does give the table name.

p. 6, l. 158   Give a reference for SMOKE.

p. 6, l. 169   Most of the input files discussed in this paper are emissions input files. Would it be helpful to insert a modifier in this sentence, such as geographic input file or regional input file?

p. 6, l. 179   "in the input file"

p. 6, l. 182   "regional expanse" – use an alternate term?

p. 8, l. 224   "i.e." –> "e.g."? (there are other emission inventories in the world)

p. 8, l. 228-229 As written there seems to be some confusion between whether the AIRCRAFT stream (one example) or the WILDFIRE stream (second example) is being discussed.

p. 9, l. 257   "aspect" –> "application"?

p. 11, l. 321-332   The first mention of most of this material is here in the Conclusions section. Would it fit better in the Introduction?

References. There are a number of missing references: and Kelly et al. (2019); Lu et al. (2020); Qin et al. (2020); Robinson et al. (2007); USEPA (2019b); Winijkul et al. (2015).

[Figure]

p. 17, Table 1   Related to Specific Comment 3 above, for the definition of the 'a' operator in the "Op" field, would adding the following parenthetical clause provide more clarity: "(which could result in the modification of an existing scaling factor)"? And some of the matching features are listed; for clarity and completeness, you could list all of them here?

p. 23, Table 7   "BNZ" –> "BENZ"?

p. 24, Table 9   Add footnote to define "Acc"?

p. 26, Figure 2, 2nd column, 2nd box from top   "Read multiple offline point «gas» emissions;"?

Section S2. This 6-page section seems a bit unnecessary, since so far as I could tell it differs from Section S1 only in the insertion of three lines (l. 371-373). An alternative could be to list just these lines in this section and state where they would be inserted in the Section S1 ECI table.

---

## Author Comment (AC2) · 26 Feb 2021

Please see attached pdf for responses to Editor and both Reviewers.

Please also note the supplement to this comment:
https://gmd.copernicus.org/preprints/gmd-2020-361/gmd-2020-361-AC2-supplement.pdf
* * *
[Figure]

2020.

---

## Author Response (AR1)

**The Detailed Emissions Scaling, Isolation, and Diagnostic (DESID) module in the Community Multiscale Air Quality (CMAQ) Modeling System version 5.3**

Benjamin N. Murphy[1], Christopher G. Nolte[1], Fahim Sidi[1], Jesse O. Bash[1], K. Wyat Appel[1], Carey Jang[2], Daiwen Kang[1], James Kelly[2], Rohit Mathur[1], Sergey Napelenok[1], George Pouliot[1] and Havala O. T. Pye[1]

[1]Center for Environmental Measurement and Modeling, U.S. Environmental Protection Agency, Research Triangle Park, North Carolina, 27711, U.S.A.
[2]Office of Air Quality Planning and Standards, U.S. Environmental Protection Agency, Research Triangle Park, North Carolina, 27711, U.S.A.

*Correspondence to*: Benjamin N. Murphy (murphy.ben@epa.gov)

**Responses to Reviewers**

Reviewer comments in black
Responses in blue

**Executive Editor Comment**

Dear authors,
in my role as Executive editor of GMD, I would like to bring to your attention our Editorial version 1.2:
https://www.geosci-model-dev.net/12/2215/2019/
This highlights some requirements of papers published in GMD, which is also available on the GMD website in the 'Manuscript Types' section: http://www.geoscientific-modeldevelopment. net/submission/manuscript_types.html In particular, please note that for your paper, the following requirement has not been met in the Discussions paper:

• Code must be published on a persistent public archive with a unique identifier for the exact model version described in the paper or uploaded to the supplement, unless this is impossible for reasons beyond the control of authors. All papers must include a section, at the end of the paper, entitled "Code availability". Here, either instructions for obtaining the code, or the reasons why the code is not available should be clearly stated. It is preferred for the code to be uploaded as a supplement or to be made available at a data repository with an associated DOI (digital object identifier) for the exact model version described in the paper. Alternatively, for established models, there may be an existing means of accessing the code through a particular system. In this case, there must exist a means of permanently accessing the precise model version described in the paper. In some cases, authors may prefer to put models on their own website, or to act as a point of contact for obtaining the code. Given the impermanence of websites and email addresses, this is not encouraged, and authors should consider improving the availability with a more permanent arrangement. Making code available through personal websites or via email contact to the authors is not sufficient. After the paper is accepted the model archive should be updated to include a link to the GMD paper.

As GitHub is not a persistent archive, please provide a persistent release for the exact source code version used for the publication in this paper. As explained in https://www.geoscientific-model-development.net/about/manuscript_types.html the preferred reference to this release is through the use of a DOI which then can be cited in the paper. For projects in GitHub a DOI for a released code version can easily be created using Zenodo, see https://guides.github.com/activities/citable-code/ for details. Finally note, that according to our new Editorial (v1.2) all data and analysis / plotting scripts should be made available.

In compliance with this request, we have added the following statement including a link to a Zenodo archive of the precise version of CMAQ code used for this study to the Code Availability Section:

"Although DESID is available in version 5.3 and later, the most recent version 5.3.2 is the default recommendation and is the version of CMAQ used for this study (https://doi.org/10.5281/zenodo.4081737)."

**Responses to Reviewer 1**

This paper describes a new feature of tracking and modifying emission inputs in the CMAQ v5.3. It is delightful to read the well-organized manuscript, with clear explanation of the workflow and functionality of DESID and its implementation in the CMAQ model. The new development is a desired tool that will make it easier for researchers and regulatory model users who are not an emission expert to study sophisticated emission perturbation simulations to quantify source-receptor relationships. This manuscript provides a detailed description of the new feature that can benefit the CMAQ user community. Therefore, I recommend the publication of this manuscript, with some suggestions to further improve the usability of this tool.

We thank the reviewer for their favorable summary of our manuscript and recommendation for publication. Detailed responses to their feedback follow.

1) While the description is comprehensive, it will be useful for the authors to provide some use cases with figures showing the emission changes or CMAQ output changes.

We provided several use cases along with figures demonstrating the emissions differences and the impact on CMAQ output concentrations in the original manuscript. For example, Fig. S1 depicts the results of example 13 from Table 5 which isolates $NO_2$ emissions from the state of Illinois. Additionally, Figs. S2 and S3 depict the results of several examples involving lumped toluene species and NO emissions outlined in Table 6.

2) It should be pointed out the using some of the DESID features will require generating and retaining sector-level emission files and all more space to run the model, although most of these files are two dimensional.

In addition to the note on page 6 line 167-168, we have added the following sentence to section 2.1:

"To overcome these and other limitations, DESID makes use of a series of generalized subroutines developed to handle critical processing steps like emission rate retrieval, error checking (e.g. for negative values), size distribution allocation, and unit conversions of all emissions streams (Fig. 3). With this uniform approach in place, model users and developers can be confident that emissions are treated as expected across all streams. If sector-specific streams are provided for 2D input (e.g. onroad and nonroad vehicles, residential wood burning, volatile chemical products) rather than one merged 2D input file, then DESID may be used to modify those specific emission sources. Although this requires more disk space to store the data needed to drive CMAQ, for many applications the added flexibility justifies the increased storage cost. Several features that accommodate common emissions processing tasks and alleviate workflow bottlenecks for research and regulatory applications build upon this robust system."

3) Figure 1 shows the workflow of DESID in which both natural and anthropogenic source s

Need clarification of question or critique.

4) Future direction: All inline emission processes, including sea-salt, dust, biogenic and plume rise, are included in the VDIFF subroutine. As the addition of DESID, the emission module may be too complicated to be treated as a sub-subroutine. Has the team considered having emission as a separate subroutine in the science process file?

CMAQ's original science formulation involved treating emissions as a boundary condition within the vertical diffusion operator. Due to the fast timescale of vertical mixing in the atmosphere, it is advantageous to include emissions in this way
because they will often dilute in the boundary layer (or above) quite quickly. Treating emissions as its own operator could lead to overprediction of chemical reaction rates or aerosol microphysical processes if pollutants are artificially concentrated in the first model layer. Please see Chapter 6 of the CMAQ science document (Byun et al., 1999) for a thorough derivation of the mathematical approach.

**Response to Reviewer 2**

**General Comments**

Pollutant emissions are a key input of chemical transport models, and many applications of these models require emissions modifications. The preparation and perturbation of input emissions is thus an important task for the use of chemical transport models, but this task is usually performed using off-line emissions processing systems. The successful use of emissions processing systems such as SMOKE, however, requires a high level of expertise and experience and typically requires time-
consuming, very detailed work to be performed in which it is easy to make mistakes and hard to track and document the process and the results. In addition, many chemical transport models now also contain in-line calculation of natural emissions that are influenced by hourly meteorology; these in-line emissions are difficult both to track by the user and to modify without code changes.

This paper describes DESID, a new module for the CMAQ chemical transport model. DESID is in essence a versatile but
powerful post-processor for off-line emissions processing systems as it enables further manipulation of the emissions fields produced by an emissions processing system. The DESID module has been designed with a very flexible, compact, yet powerful user interface. Its use can save considerable emissions preparation time and reduce computer storage requirements while providing control over different emissions streams, error checking, and useful output documentation as well as optional output of processed emissions files. The authors correctly note that resources are seldom available for "model
refactoring and infrastructure development", but this new module is a welcome exception to this usual constraint and it constitutes a significant addition to the well-established CMAQ air quality modeling system. It will likely be adopted quickly and widely by the large CMAQ user community. I found this to be a well-structured and well-written paper. I recommend its acceptance after minor revision. To this end I have made a number of specific comments and suggestions below that I believe will improve the final version and that I hope the authors will consider.

We sincerely thank the reviewer for their gracious comments about the potential impact of DESID's features for the CMAQ user community. Below we address the thoughtful suggestions the reviewer has provided and thank them for contributing to the quality of the final manuscript.

**Specific Comments**

1. It is mentioned on page 2 that emissions processing must address temporal allocation, but the time variation of input
emissions files and how DESID treats this variation is only mentioned once in the paper, on page 5, line 132, and I am not sure this mention is totally correct. Input emissions are commonly specified for each hour of the day. If the model time step is less than one hour, will DESID still apply the instruction rules during each model time step as stated. This may be true for online emissions streams, but is it also true for offline emissions streams? A few words about how the DESID emissions module fits into the CMAQ operator sequence would be helpful as well as a mention of the day-to-day variation of offline
emissions streams (i.e., day-specific or day-of-week specific and monthly or seasonal variation).

The phrasing on page 5, line 32 is correct but is not very clear. As the text states, we constructed the DESID algorithm to process the user-defined scaling rules during the initialization step. The results of this processing are stored in a series of arrays and applied uniformly to the emissions calculated at every model time step (i.e. synchronization step, ~5 mins for a 12 km scale simulation). In previous and current CMAQ versions, the emissions are read for the time points available on the input file immediately before and after the model time point (i.e. beginning and end of an hour). Emissions for the current model time point are then interpolated. For the next model time point, emissions are only read again if the model time has surpassed the range of the data already read in (i.e. advanced to the next hour). Otherwise, the emissions are interpolated again from the data that has already been read.

For DESID, we opt to apply the scaling rules after the interpolation has been performed rather than applying it to the raw data that is read in from the input files. Although this approach is technically less efficient for most cases (e.g. assuming twelve 5 minute time steps per hour for a typical simulation, there are perhaps 10 or 11 redundant multiplication steps), we found that:

-   The runtime penalty for these extra multiplication steps is not a bottleneck for overall CMAQ resources relative to other model processes.
-   The CMAQ input data interpolation routines are narrowly focused and editing those lower-level functions or creating an additional layer of modified raw data would likely obfuscate data flows and make debugging more difficult, especially for external users.
-   Applying scaling to the time-specific emissions preserves one of the core concepts of DESID, treating online and offline emission streams as similarly as possible in the coding structure to minimize ongoing maintenance.

We have revised the first paragraph of section 2.2.2 to the following:

"The *Emissions Scaling* component is formatted as a table with a user-defined number of rows, each corresponding to an individual rule. These rules may be logistically simple (e.g. map the variable named NO from one emissions stream directly to the CMAQ species NO) or considerably complex (e.g. scale multiple species by 75% that have already been mapped for all emission streams). During the CMAQ initialization process, DESID reads these rules and translates them into a series of low-level instructions that are stored in several persistent arrays. These arrays are then applied uniformly in time to the base emissions after calculation (for online emissions) or interpolation (for offline emissions).to be carried out during each model time step. Each instruction involves at most one emission stream, one CMAQ variable and one emission variable. The simplest rules usually translate to one instruction, but the more complex ones (e.g. affecting multiple streams or multiple CMAQ species) are made up of several instructions.

We have also added some explanation at the end of section 2.2.2 to further address temporal variability of various scales:

"As stated earlier, DESID scaling rules are applied uniformly in time and there is currently not an ability to redistribute emissions in time (e.g. modify the diel profile of a stream or species). DESID is unaware of any hourly or daily variability in offline emissions, so this feature should be captured using upstream emissions processing tools. By default, DEISD will generate an error if it finds that the model simulation day does not match the day defined on an offline emission input file. It is common for emissions platforms to use representative days for particular offline streams (e.g. weekend-weekday, weekly, monthly, seasonal). In these cases, the date-matching requirement in DESID may be overridden by setting the environment variables of the format GR_EM_SYM_DATE_xxx and STK_EM_SYM_DATE_yyy to true."

Finally, we have added a sentence to the end of the first paragraph of section 2.1 to specify where emissions are handled in the science process operator flow.

"Once emissions are calculated they are introduced to the model atmosphere as part of the solution for vertical diffusion, which is the first operator solved during the model synchronization time step (Byun et al., 1999)."

2. It is also mentioned on page 2 that emissions processing must address chemical speciation, but in the discussion of the Emissions Scaling rules and Table 1 in Section 2.2.2, it is not made clear that one of the eight rule fields, "Emission Variable", is not an inventory emission variable but rather is a speciated emission variable output by the emissions processing system.

We have added the following sentence to Section 2.2.2:

"Each rule is articulated with 8 fields (**Error! Reference source not found.**). Examples 1 and 2 in **Error! Reference source not found.** demonstrate rules to map NO and fine-mode elemental carbon (EC), respectively, for all emission streams. In typical cases, the emission variables in these examples will be populated by an upstream emission processor during a chemical speciation step that converts emission inventory pollutants to model-relevant species. A broader emission inventory variable like total volatile organic compounds (VOCs) or particulate matter
(PM) may be used if it is available on the emission stream. In that case, any scaling rules would apply uniformly to all sources that contribute to that emission stream. More complicated scaling rules are possible by exercising options in the available fields."

3. I found the use of examples in Section 2 very helpful for understanding how DESID works. One point that was not clear to me, however, appears in lines 139-141. If two rules apply to the same region, stream, and emission variable, would this
result in the second rule modifying the scaling factor of the instruction corresponding to the first rule? That is, can multiple rules sometimes result in a smaller number of instructions?

No, although there are cases like the one presented where you could imagine combing scaling rules into fewer instructions, we did not provide the logic to simplify this step. We may implement this in the future but there are at least two reasons why we may not: 1) we want to preserve the user-supplied directives for output to the diagnostic log so they can check that
DESID is interpreting rules correctly and 2) this would only be applicable when the most recent rule for a stream, emission variable, and model species was applied over the same region (or subset) and used the same scaling basis. The use case seems relatively rare, and so we opt to ignore it for now.

4. Section 2.2 does not say what is the minimum number of rules required by the DESID ECI. Can it be zero (i.e., no rules), or if not could the minimum number of rules be given in a new table in the Supplement?

If the user supplies zero rules, then DESID will implement no emissions and CMAQ will run with initial and boundary condition sources. All of the offline and online emissions will still be interpolated and calculated, respectively, but none will be added to the DESID output emission array. We have specified this in the beginning of section 2.2 as follows:

"Each instruction involves at most one emission stream, one CMAQ variable and one emission variable. The simplest rules usually translate to one instruction, but the more complex ones (e.g. affecting multiple streams or multiple
CMAQ species) are made up of several instructions. If no rules are provided to DESID or an ECI is not specified, then CMAQ will introduce no emissions to the model."

5. An important constraint on DESID noted on line 167 is the degree of disaggregation of the emissions streams input by the model. In lines 340-341 the authors note a traditional treatment of emissions with 2 offline gridded and 8 offline point emissions files vs. a much more detailed treatment with 27 offline gridded and 19 offline point emissions files. The authors
might consider adding a new table to the Supplement that lists and compares these two sets of emissions streams (with corresponding environment variables?) as an example of a simulation that makes full use of the capabilities of DESID and what this higher level of inventory disaggregation looks like.

We have added Section S4 and Table S1 which, respectively, give examples of the CMAQ runscript for the two cases and present the streams considered for reference.

6. I am concerned that the use of Regional Definitions, while very intuitive and appealing, might not always give exactly the expected results when applied to multiple emissions streams. The issue is how to handle grid cells that straddle two or more regions. The authors state on lines 172-173 that the gridded definition of a region should quantify "the fraction of emissions in each model grid cell that is associated with the region", but I would expect that the value of the fraction of emissions for a border grid cell would frequently vary with the emissions stream. For example, consider the spatial distribution of ammonia emissions from on-road motor vehicles for a grid cell straddling the Indiana-Illinois state line, which will depend on road network geometry, vs. ammonia emissions from fertilizer application, which will depend on the distribution of farms. In effect I think applying a Regional Definition is equivalent to applying the same spatial surrogate to all emissions streams. I definitely am not saying that this limitation should preclude the use of Regional Definitions, but if the authors agree with this concern, then it would be worth noting in the manuscript that some uncertainty will inevitably be associated with the apportionment of emissions between neighboring regions due to this issue. This minor caveat is particularly important for policy applications, where different fractional reductions may be under consideration for different jurisdictions but the actual modeled reductions may be slightly different. Also, I expect this relative uncertainty would likely increase as the size of a region decreases (Rhode Island, say, or a smaller city).

This a critically important point for many applications and we appreciate the reviewer reminding us to mention it explicitly. As the reviewer implies, the region-based scaling available in DESID is meant for either a) applications with large regions where uncertainties along borders are swamped by large overall changes, or b) highly experimental scenarios (think back of the envelope) to help determine whether or not a hypothesis warrants a detailed offline spatial allocation task.

For cases where a user really wants to segregate the emissions of small areas, for example, for source apportionment or sensitivity purposes, they are best off either a) providing those emissions via independent streams or b) providing a region mask that is reflective of the actual spatial allocation of their emissions of interest. We have added the following paragraph to the end of section 2.2.3:

"Although DESID's region-based scaling capability is useful for many applications, it can introduce potentially important uncertainties when the scale of the model grid is insufficient for capturing the distribution of pollutants between two neighboring boundaries. For example, consider a region mask specifying the domain of Illinois including real fractions that are area weighted. If a hypothetical border grid cell contains far more Illinois emissions from some sector than the area-weighted fraction would indicate due to the distribution of population, road networks, or farmland, etc., then errors will be introduced by applying the area-weighted fractions during scaling. If these errors must be avoided, users are advised to provide region masks that are reflective of a more appropriate weighting or provide emission streams segregated by the regions they wish to modify."

**Technical Corrections/Suggestions**

p. 2, l. 32 Perhaps "... managers in designing programs to improve urban- ..."

Accepted.

p. 2, l. 43 It is not clear here whether "sources" refers to individual facilities or to different source sectors or source types.

Added, "…thousands of sources, including individual facilities and distributed activities."

p. 3, l. 71 It is noted here that the paper describes the DESID module as it exists in CMAQ version 5.3.2. Should the paper title reflect this version number?

We have modified the title to reflect version 5.3.2 and mentioned in the last paragraph of the introduction the main features that were added between v5.3 and v5.3.2.

p. 3, l.77 Perhaps "... and online gridded (e.g., biogenic and marine vapors, ..."

Accepted.

p. 4, l. 103 Perhaps "The rest of this section demonstrates ..."

Accepted.

p. 5, 2nd paragraph. Change the order of the third and fourth sentences?

Accepted.

p. 5, l. 149 Should it be "map emissions variables to CMAQ species with differing units"?

Accepted.

p. 6, l. 157 Just to comment that the table name is not given here but a similar description on l. 221 does give the table name.

Added the table name EMIS_SURR_TABLE.

p. 6, l. 158 Give a reference for SMOKE.

Done.

p. 6, l. 169 Most of the input files discussed in this paper are emissions input files.

Would it be helpful to insert a modifier in this sentence, such as geographic input file or regional input file?

We assume the reviewer is referring to the first sentence of section 2.2.3. We have inserted 'geographic' in the sentence as
suggested.

p. 6, l. 179 "in the input file"

Accepted.

p. 6, l. 182 "regional expanse" – use an alternate term?

We have revised the sentence as follows: "broader U.S. geographical area"

p. 8, l. 224 "i.e." –> "e.g."? (there are other emission inventories in the world)

Accepted.

p. 8, l. 228-229 As written there seems to be some confusion between whether the AIRCRAFT stream (one example) or the WILDFIRE stream (second example) is being discussed.

Revised to read:

"These default parameters may be overridden at the stream level by subsequent entries though, as shown for the AIRCRAFT stream in **Error! Reference source not found.**. Following the AIRCRAFT specification, WILDFIRE aerosol parameters are set with wildfire specific labels (WILD_FINE and WILD_COARSE) rather than using the existing labels FINE and COARSE."

p. 9, l. 257 "aspect" –> "application"?

Accepted.

p. 11, l. 321-332 The first mention of most of this material is here in the Conclusions section. Would it fit better in the Introduction?

Good suggestion. Moved.

References. There are a number of missing references: and Kelly et al. (2019); Lu et al. (2020); Qin et al. (2020); Robinson
et al. (2007); USEPA (2019b); Winijkul et al. (2015).

Fixed.

p. 17, Table 1 Related to Specific Comment 3 above, for the definition of the 'a' operator in the "Op" field, would adding the following parenthetical clause provide more clarity: "(which could result in the modification of an existing scaling factor)"? And some of the matching features are listed; for clarity and completeness, you could list all of them here?

Accepted changes to Table 1.

p. 23, Table 7 "BNZ" –> "BENZ"?

Corrected.

p. 24, Table 9 Add footnote to define "Acc"?

Updated p. 26, Figure 2, 2nd column, 2nd box from top "Read multiple offline point «gas» emissions;"?

Updated.

Section S2. This 6-page section seems a bit unnecessary, since so far as I could tell it differs from Section S1 only in the insertion of three lines (l. 371-373). An alternative could be to list just these lines in this section and state where they would be inserted in the Section S1 ECI table.

Accepted.

**References**

Byun, D. W. and Young, J. and Odman, M. T., 1999, "Governing equations and computational structure of the Community Multiscale Air Quality (CMAQ) chemical transport model," *Science Algorithms of the EPA models-3 Community Multiscale*

*Air Quality (CMAQ) Modeling System,* National Exposure Research Laboratory, U.S. EPA, Research Triangle Park, N.C., Chap. 6, https://www.cmascenter.org/cmaq/science_documentation/pdf/ch06.pdf.